# Reduced Expression of Annexin A6 Induces Metabolic Reprogramming That Favors Rapid Fatty Acid Oxidation in Triple-Negative Breast Cancer Cells

**DOI:** 10.3390/cancers14051108

**Published:** 2022-02-22

**Authors:** Stephen D. Williams, Amos M. Sakwe

**Affiliations:** Department of Biochemistry, Cancer Biology, Neuroscience and Pharmacology, School of Graduate Studies and Research, Meharry Medical College, Nashville, TN 37208, USA; swilliams17@email.mmc.edu

**Keywords:** Annexin A6, breast cancer, metabolic reprogramming, lipid metabolism, tyrosine kinase inhibitors

## Abstract

**Simple Summary:**

The expression status of Annexin A6 (AnxA6) has been shown to influence tumor growth, metastasis, and resistance to therapeutic intervention, but the contribution of this tumor suppressor in the metabolic adaptation of basal-like (AnxA6-low) versus mesenchymal-like (AnxA6-high) TNBC subsets remains unclear. The downregulation of AnxA6 in TNBC cells attenuated mitochondrial respiration, glycolytic flux, and cellular ATP production capacity, resulting in a quiescent metabolic phenotype. The overexpression or chronic lapatinib induced the expression of AnxA6 in AnxA6-low TNBC cells and reversed the quiescent phenotype to a more lipogenic/glycolytic phenotype. Interestingly, AnxA6-depletion was associated with rapid fatty acid uptake and oxidation (lypolytic metabolic phenotype) but decreased lipid droplet accumulation. Our data suggest that the expression status of AnxA6 in TNBC cells is associated with distinct metabolic adaptations of basal-like and mesenchymal-like TNBC subsets in response to cellular stress and/or therapeutic intervention.

**Abstract:**

The ability of cancer cells to alter their metabolism is one of the major mechanisms underlying rapid tumor progression and/or therapeutic resistance in solid tumors, including the hard-to-treat triple-negative breast cancer (TNBC) subtype. Here, we assessed the contribution of the tumor suppressor, Annexin A6 (AnxA6), in the metabolic adaptation of basal-like (AnxA6-low) versus mesenchymal-like (AnxA6-high), as well as in lapatinib-resistant TNBC cells. Using model basal-like and mesenchymal-like TNBC cell lines, we show that TNBC cells also exhibit metabolic heterogeneity. The downregulation of AnxA6 in TNBC cells generally attenuated mitochondrial respiration, glycolytic flux, and cellular ATP production capacity resulting in a quiescent metabolic phenotype. We also show that AnxA6 depletion in mesenchymal-like TNBC cells was associated with a rapid uptake and mitochondrial fatty acid oxidation and diminished lipid droplet accumulation and altered the lipogenic metabolic phenotype of these cells to a lypolytic metabolic phenotype. The overexpression or chronic lapatinib-induced upregulation of AnxA6 in AnxA6-low TNBC cells reversed the quiescent/lypolytic phenotype to a more lipogenic/glycolytic phenotype with gluconeogenic precursors as additional metabolites. Collectively, these data suggest that the expression status of AnxA6 in TNBC cells underlies distinct metabolic adaptations of basal-like and mesenchymal-like TNBC subsets in response to cellular stress and/or therapeutic intervention and suggest AnxA6 as a biomarker for metabolic subtyping of TNBC subsets.

## 1. Introduction

Triple negative breast cancer (TNBC), which accounts for up to 17% of all breast cancer cases in the United States [1], disproportionately affects premenopausal African American and Hispanic women [1,2,3,4,5,6,7]. The lack of hormone receptors and HER2 expression and, therefore, the inability to use approved hormone receptor-targeted therapeutic options makes chemotherapy the standard of care (SOC) for patients with the TNBC subtype [8]. While proven beneficial for patients with early-stage disease, patients that were diagnosed with late-stage disease display poor therapeutic responses [1,8,9]. Although up to 70% of TNBCs express amplified levels of epidermal growth factor receptor (EGFR), targeting the receptor with tyrosine kinase inhibitors (TKIs) and/or therapeutic monoclonal antibodies even in combination with chemotherapy [10,11,12,13] has also been met with dismal patient responses and frequent relapse and metastasis [14,15,16,17]. This poor therapeutic response of TNBC may be due to the well described molecular heterogeneity of the TNBC tumors and phenotypic diversity that is depicted by the existence of basal-like (BSL) and mesenchymal-like (MES) TNBC subsets [18,19,20,21,22,23].

After significant efforts on TNBC molecular classifications and the associated predictions of prognosis and response to therapy, evidence linking the distinct metabolic dependencies of TNBCs with molecular subtypes and clinical outcomes is beginning to emerge [19,24]. However, Lanning et.al., demonstrated that the BSL-TNBC cell lines HCC-70 and MDA-468 were metabolically more active compared to the MES-TNBC cell lines MDA-231 and Hs578t [25]. While this is among the early studies describing the metabolic heterogeneity of TNBC cells, the significantly lower OXPHOS capacity of MES-TNBC cell lines has been reported to underlie the adaptive response of these cells to receptor tyrosine kinase (RTK) inhibition and other metabolic perturbations [25]. However, our understanding of the metabolic dependencies of the more aggressive BSL-TNBCs remains limited.

In recent years, the Ca^2+^-dependent membrane-binding tumor suppressor, Annexin A6 (AnxA6) has been shown to be implicated in a wide range of cellular functions including cell growth, differentiation, motility, cell surface-mediated signaling, cholesterol homeostasis, Ca^2+^ homeostasis, energy metabolism, and membrane repair [26,27,28]. However, the involvement of AnxA6 in cellular energy needs is an emerging notion and remains poorly understood. Fibroblasts that were isolated from AnxA6^−/−^ mice demonstrated fragmented mitochondria, impaired respiration, and resistance to Ca^2+^-mediated apoptosis [29]. Recent studies also show that the expression of AnxA6 in adipocytes underlies their ability to impair fat storage and adiponectin release during oxidative stress [30], and that the lack of AnxA6 in hepatocytes compromises alanine-dependent gluconeogenesis and liver regeneration in mice [31]. Although showing a vital role in the metabolic vulnerabilities of adipocytes, hepatocytes, and the associated chronic diseases, it remains unclear whether AnxA6 plays a critical role in the metabolic reprogramming of the phenotypically distinct TNBC subtypes and as reported recently following acquired resistance to the highly potent but poorly efficacious EGFR-TKIs [32].

Recent studies on AnxA6 and TNBC biology have provided evidence suggesting that the expression status of AnxA6 can be used to delineate basal-like (AnxA6-low) from mesenchymal-like (AnxA6-high) TNBCs cells and patient-derived xenograft models [33]. Furthermore, the reduced expression of AnxA6 has been shown to promote rapid tumor growth and affect several aspects of energy metabolism including adipogenesis [30,34]), and gluconeogenesis [31]. Interestingly, chronic treatment of AnxA6-low BSL-TNBC cell lines with EGFR-TKIs but not chemotherapy, led to AnxA6 upregulation and the accumulation of cholesterol in late endosomes, suggestive of a novel mechanism of TKI resistance [32]. Although the expression status of AnxA6 has been shown to influence tumor growth, metastasis and resistance to therapeutic intervention, the contribution of this tumor suppressor in the metabolic adaptation of TNBC cells remains poorly understood. Our data not only suggest that the reduced expression of AnxA6 is accompanied by metabolic reprogramming in favor of fatty acid oxidation and distinct bioenergetic adaptations to stress. Furthermore, we show that basal-like (AnxA6-low) and mesenchymal-like (AnxA6-high) TNBCs cells exhibit distinct metabolic phenotypes, suggesting that AnxA6 expression status can be used in the future as a metabolic biomarker for the subtyping of TNBC subsets.

## 2. Materials and Methods

### 2.1. Cell Culture

BT-549 and MDA-MB-231, (mesenchymal-like) and MDA-MB-468 and HCC70 (basal-like) breast cancer cell lines were purchased from American Type Culture Collection (ATCC). The cells were expanded, cryopreserved at −80 °C, and only early passages (<passage 5) of these cell lines were used in the experiments that were described in this manuscript. The prophylactic mycoplasma treatment of the cells was also routinely carried out on recovery of the cell stocks. BT-549, HCC70, and MDA-MB-231 cell lines were cultured in DMEM/F12 containing 10% fetal bovine serum and Pen/Strep (100 units/mL penicillin and 50 units/mL streptomycin). MDA-MB-468 cells were maintained in medium L-15 (Leibovitz) containing 10% fetal bovine serum, Pen/Strep, and 0.15% sodium bicarbonate. The cells were maintained at 37 °C in a humidified CO_2_ incubator and sub-cultured by trypsinization using 0.25% trypsin/0.53 mM EDTA solution (Invitrogen, Carlsbad, CA, USA). Where indicated, serum-starvation of cells was achieved by culturing the cells overnight in their respective media containing 0.5% fetal bovine serum. For treatment of the cells with drugs, the cells were seeded in either 10-cm dishes or 96-well plates and allowed to attach overnight in complete medium. The medium was aspirated and replaced with fresh medium containing the control dimethyl-sulfoxide (DMSO), or the indicated concentrations of the indicated drug. The drug-containing media were replaced every 2 days.

### 2.2. Antibodies and Other Reagents

Antibodies against AnxA6, as well as secondary anti-mouse, anti-goat, and anti-rabbit horseradish peroxidase-conjugated antibodies were purchased from Santa Cruz Biotechnology (Dallas, TX, USA). The antibodies against COXIV, Cytc. C, Pyruvate dehydrogenase (PDH), and Superoxide dismustase (SOD1) were purchased from Cell Signaling Technology (Beverly, MA, USA). Oil Red-O, Rotenone, 2-Deoxy-D-Glucose, Etomoxir, and antibody against β-actin (ACTB) were purchased from Sigma Aldrich (St. Louis, MO, USA). A set of EGFR-TKIs including lapatinib-ditosylate, erlotinib, gefitinib, and canertinib were purchased from BioVision (Milpitas, CA, USA). The Mito Stress Test Assay, Palmitate FAO Assay, and the culture media were purchased from Agilent Technologies (Santa Clara, CA, USA). Except where otherwise indicated, all the other reagents were purchased from Sigma Aldrich and/or Cell Signaling Technology.

### 2.3. Plasmid Constructs and Transfections

BT-549 and MDA-468 breast cancer cell lines that were stably transfected with non-silencing shRNA (NSC) or AnxA6-targeting shRNAs (A6sh5), respectively, were generated and validated as previously described [35]. The MDA-468 cells overexpressing flag-tagged AnxA6 were generated as recently reported [34]. The lapatinib-resistant (Lap-R) MDA-468 cell lines that were stably transfected with the NSC and A6sh5 shRNAs were established and maintained as previously described [16,17,24]. AnxA6 protein expression was verified by immunoblotting.

### 2.4. Lipid Droplet Analysis

The cells (1 × 10^6^) were seeded on glass cover slips in 6-well plates and incubated for 24 h at 37 °C in a humidified CO_2_ incubator. The cells were then fixed using 4% formaldehyde (PFA) made in PBS at room temperature for 5 min, and washed once using 60% isopropyl alcohol for 5 min. Oil Red-O stock solution (0.25%) (Sigma-Aldrich, St. Louis, MO, USA) was prepared in 20 mL of isopropyl alcohol and diluted in distilled water and filtered before use. The diluted Oil Red-O solution was used to stain the cells at room temperature for 20 min, followed by rinsing with distilled water, and followed by staining with hematoxylin (Sigma-Aldrich) to visualize nuclei. The cells were imaged on a Keyence BZ-X800 fluorescence microscope (Keyence Corp., Osaka, Japan). A total of four fields of each condition were used for the quantification of the Oil red-O staining using the NIH ImageJ software.

### 2.5. Western Blotting

The cells were seeded in 10 cm dishes and cultured until they were 70% confluent. The cells were scraped in ice-cold PBS and whole-cell lysates were prepared in radioimmune precipitation assay (RIPA) buffer (50 mm Tris-HCl, pH 7.4, 1% Nonidet P-40, 0.1% sodium deoxycholate, 150 mm NaCl, 1 mm EDTA) containing protease inhibitor cocktail (Sigma) and phosphatase inhibitors (20 mm sodium fluoride, 50 mm β-glycerophosphate, and 1 mm sodium orthovanadate). The cleared cell lysates were separated in 4–12% SDS-polyacrylamide gels and then transferred to nitrocellulose membranes. The membranes were subsequently probed with mouse anti-AnxA6 antibody (1:3000 dilution). The detection of β-Actin (1:10,000) was used as the loading control. The blots were revealed by enhanced chemiluminescence (Perkin Elmer, Waltham, MA, USA), scanned, and quantified using NIH Image J software (Version 1.53o, Bethesda, MD, USA). All original western blots are included in Appendix A.

### 2.6. Isolation of Mitochondria from TNBC Cells

The cells were seeded in two 15 cm dishes and cultured until 80–90% confluence. Crude preparations of the mitochondria were carried out as previously described [36]. Briefly, the cells were washed twice and harvested by scrapping in PBS. The cell pellet was then resuspended in ice-cold hypotonic cell resuspension (RSB) buffer (10 mM NaCl, 10 mM Tris-HCl pH 7.5, and either 1.5 mM MgCl_2_ or 1.5 mM CaCl_2_), incubated on ice for 10 min, and Dounce homogenized. An equal volume of 2.5× mitochondria solubilization (MS) buffer (175 mM sucrose, 12.5 mM Tris-HCl, 2.5 mM EDTA pH 7.5) was immediately added and the homogenates were centrifuged at 1300× *g* for 5 min at 4 °C. The supernatant was transferred to new tubes and centrifuged as above for a total of three times. The final 1300× *g* supernatant was then centrifuged at 15,000× *g* for 15 min at 4 °C. The post-mitochondria supernatant (PMS) was transferred to new tubes, while the mitochondria-enriched pellet (Mito) was washed twice with 1× MS buffer by centrifugation at 15,000× *g* for 15 min at 4 °C. The washed mitochondria pellet was resuspended in RIPA buffer containing protease inhibitor cocktail (Sigma), incubated on ice for 30 min, and centrifuged at 15,000× *g* to obtain soluble mitochondria proteins for Western blotting.

### 2.7. Cell Viability Assay

The cells were seeded in 96-well plates overnight in triplicates using 5.0 × 10^4^ cells/well for the MDA-468 cell lines (NSC, A6sh5, and Lap-R), and 2.5 × 10^4^ cells/well for the BT-549 cell lines (NSC, A6sh5, and Lap-R). After 72 h incubation period at 37 °C in a humidified CO_2_ incubator, proliferation and viability of the cells were determined using the Prestoblue reagent (Invitrogen) diluted 1:10 in serum-free medium according to the manufacturer’s instructions. The cells were incubated for 2 h at 37 °C in a humidified CO_2_ incubator and the fluorescence was measured on a Synergy HT multidetection microplate reader (BioTek, Winooski, VT, USA) at Ex/Em wavelengths 535/590.

### 2.8. Analysis of Mitochondria Respiration in TNBC Cells Using the Seahorse XF Analyzer

The oxygen consumption rate (OCR) and the extracellular acidification rate (ECAR) were measured by mitochondria stress test using an XFe96 extracellular flux analyzer (Seahorse Bioscience, North Billerica, MA, USA), as previously described [25]. Briefly, 5.0 × 10^4^ cells were seeded in XFe96-well plates overnight in respective complete media, and the OCR and ECAR were measured in Seahorse XF DMEM pH 7.4 (Agilent Technologies) that was supplemented with 10 mM D-glucose, 2 mM L-glutamine, and 1 mM sodium pyruvate. As indicated, the cells were treated with the following pharmacological modulators of mitochondrial respiratory chain proteins: 1.5 µM oligomycin, 2.0 µM carbonyl cyanide p-tri-fluoromethoxyphenylhydrazone (FCCP), 0.5 µM antimycin A, and 0.5 µM rotenone (Seahorse bioscience). A total of three basal rate measurements were taken (0–14 min) prior to injection of the compounds, followed by three measurements of OCR/ECAR following the injection of each drug. OCR and ECAR readings were normalized to total protein levels (BCA protein assay, Pierce) in each well. Each cell line was seeded in 96 wells per experiment and replicate experiments were carried out at least three times.

### 2.9. Fatty Acid Uptake

The fatty acid uptake was assessed using the fatty acid uptake assay kit according to the manufacturer’s instructions (Sigma-Aldrich). Briefly, 2.5 × 10^4^ BT-549 and 5.0 × 10^4^ MDA-468 control and Lap-R cells were seeded in 96-well plates and incubated in the respective complete media for 24 h at 37 °C, 5% CO_2_. The cells were washed with PBS, serum deprived for 1 h, and then incubated with a fluorescently-labeled fatty acid substrate for 1 h at 37 °C in a humidified CO_2_ incubator. The fatty acid uptake was assessed by measuring the fluorescence signal at Ex/Em = 485/515 nm using a Synergy HT microplate reader (BioTek).

### 2.10. Palmitate Oxidation Stress

The fatty acid oxidation assay was carried out by using the Seahorse XF palmitate oxidation stress test as recommended by the manufacturer (Agilent Technologies). A total of 5.0 × 10^4^ cells/well were seeded in XFe96-well plates overnight containing complete DMEM/F12 medium, then cultured an additional 24 h in substrate-limited XF medium (0.5 mM Glucose, 1.0 mM Glutamax, 0.5 mM L-carnitine, and 1% FBS). On the day of the assay, the media was replaced with FAO medium [KHB: 111 mM NaCl, 4.7 mM KCl, 1.25 mM CaCl_2_, 2 mM MgSO_4_, and 1.2 mM sodium phosphate buffer that was supplemented with 2.5 mM glucose, 0.5 mM carnitine, and 5 mM Hepes (adjusted to pH 7.4)] for 30 min in a non-CO_2_ incubator. Half of the wells/cell line were then preincubated with Etomoxir (ETO; 100 μM final, Agilent) for 30 min and the mitochondria stress test was initiated in the presence of carrier BSA control or BSA-conjugated Palmitate (Palmitate-BSA) and performed as described in the measurement of ECAR and OCR.

### 2.11. Nuclear Magnetic Resonance (NMR)-Based Metabolite Measurement

The TNBC cell lines were seeded overnight and given complete medium the following day. The cells were cultured for 48 h and the metabolites were extracted as previously described [25] and dried by using a SpeedVac concentrator (Thermo Scientific, Waltham, MA, USA). For the NMR sample preparation, the dried samples were resuspended in phosphate buffered D_2_O-containing 0.75% trimethylsilyl propanoic acid (TSP). NMR spectra were acquired at the Vanderbilt NMR Core facility using a 14.0 T Bruker magnet that was equipped with a Bruker AV-III console operating at 600.13 MHz. All spectra were acquired in 3 mm NMR tubes using a Bruker 5 mm BBFO NMR probe. For 1D ^1^H-NMR, data were acquired using the 1D-NOE using the following experimental conditions: sample temperature of 300 K, 96 k data points, 20 ppm sweep width, a recycle delay of 4 s, a mixing time of 150 ms, and 32 scans.

### 2.12. Immunofluorescence Microscopy

The cells were seeded on cover clips in 6-well plates until 70% confluence. The cells were then fixed in 3.7% formaldehyde that was diluted in PBS for 15 min at room temperature. Formaldehyde was aspirated and the cells were rinsed gently 2× with PBS. The cells were subsequently permeabilized in PBS, 10%FBS, 0.1% triton ×100 for 30 min at room temperature, followed by incubation with primary antibodies (anti-AnxA6, 1:300; anti-COXIV, 1:250) at room temperate for 1 h. After three washes with PBS, the cells were incubated with secondary antibodies for 1 h at room temperature followed by three washes with PBS. The coverslips were then mounted with EverBrite^TM^ Mounting medium and imaged using the Nikon A1R confocal microscope (Nikon, Tokyo, Japan).

## 3. Results

### 3.1. Metabolic Adaptation of TNBC Cells Is AnxA6-Dependent

We have previously shown that the downregulation of AnxA6 in TNBC cell lines led to increased cell proliferation, reduced cell motility, rapid growth of xenograft tumors, and was associated with poor overall survival of basal-like TNBC patients [34,35]. In our quest to better understand the basis for these conspicuous differences in TNBC cellular metabolic phenotypes, we hypothesized that the expression status of AnxA6 affects cellular bioenergetics and the metabolic capacity in the phenotypically-distinct basal-like and mesenchymal-like TNBC cells. To test this, we measured OCR and ECAR in a non-silencing control (NSC) and AnxA6 downregulated (A6sh5) MDA-468 and BT-549 TNBC cells that were generated and validated previously [32,35] and depicted in Figure 1A,B. Using the mitochondrial stress test, the analysis revealed that the downregulation of AnxA6 in the AnxA6-high BT-549 cells (Figure 1C) and AnxA6-low MDA-468 (Figure 1D) surprisingly strongly suppressed not only the mitochondrial respiration, but also the ability of these cells to alter their metabolic dependency on glycolysis. In either case, the control AnxA6-expressing MDA-468 or BT-549 cells were profiled as highly energetic phenotypes, while the downregulation of AnxA6 (A6sh5) paradoxically altered the metabolic capacity of the cells to a more quiescent state (Figure 1D,E).

To confirm these observations, we next assessed the effects of AnxA6 overexpression in the AnxA6-low MDA-468 cells [35]. We first confirmed that AnxA6 expression was at least seven-fold higher in the flag-tagged AnxA6 (Flag-A6) compared to the control empty-vector (EV) transfected MDA-468 cells (Figure 2A, B). Compared to the EV control cells, the overexpression of AnxA6 in this basal-like breast cancer cell line was associated with higher OCR and ECAR (Figure 2C,D), as well as significantly higher proton leak (*p* = 0.0361) and ATP production (*p* = 0.0284) (Figure 2E). Interestingly, the overexpression of AnxA6 in this cell line led to a more energetic phenotype compared to the more quiescent MDA-EV cell line (Figure 2F). Together with the data in Figure 1, this suggests that the expression status of AnxA6 is associated with remarkable changes in the metabolic capacity of TNBC cells.

### 3.2. Downregulation of AnxA6 Disrupts Mitochondrial Function and Integrity

Based on our results that AnxA6-depletion in TNBC cells alters their metabolic capacity, we next explored the molecular basis for the remarkable switch from the highly energetic phenotype of AnxA6-expressing to the quiescent phenotype following AnxA6-depletion in TNBC cells. We first examined whether AnxA6 depletion in the TNBC cells affected the mitochondrial function and, therefore, the ATP production capacity of these cells. We isolated mitochondria-enriched fractions and the post-mitochondria supernatants from the BT-549 and MDA-468 TNBC cells as previously described [36] and assessed the expression of AnxA6 and mitochondria markers cytochrome C oxidase subunit IV (COXIV) and Cytochrome as well as the mostly cytoplasmic superoxide dismutase (SOD1) and pyruvate dehydrogenase (PDH) by Western blotting. This analysis revealed that COXIV and cytochrome c were mostly detected in the mitochondria-enriched fraction and barely detectable in the post-mitochondria supernatant (Figure 3A,B).

In this assay, PDH and SOD1 were mostly in the post-mitochondria supernatant and interestingly AnxA6 was not only associated with mitochondria in both the cell lines, but also the cell membranes in the post-mitochondria supernatant (Figure 3A). To confirm that AnxA6 is indeed targeted to the mitochondria in a Ca^2+^-dependent manner, we show that >two-fold AnxA6 was detected in the mitochondria that were isolated following incubation, cell swelling, and disruption in Ca^2+^-containing RBS buffer compared to those that were isolated in the Mg^2+^-containing RBS buffer (Figure 3C, lanes 2 and 4). To further confirm that AnxA6 is targeted to the mitochondria, we co-stained the mitochondria marker COXIV and AnxA6, and showed that AnxA6 co-localized with COXIV in TNBC cells (Figure 3E). Figure 3E also shows that mitochondria in AnxA6-expressing cells were not only more abundant but also appeared to be intact based on the contiguous staining. On the contrary, mitochondria in the AnxA6-depleted cells were fewer and appeared fragmented as revealed by the punctate staining. Consistent with this observation, proton leak and ATP production (Figure 3F) in AnxA6-expressing control BT-549-NSC cells were significantly higher (*p* = 0.006712) compared to those in AnxA6-depleted BT-549-A6sh5 cells. Similarly, the proton leak and ATP production (Figure 3G) in the AnxA6-expressing control MDA-468 were also significantly decreased (*p* = 0.000056) following AnxA6-downregulation in these cells. These data lend credence to the possibility that AnxA6 is associated with mitochondria and that mitochondria that are associated AnxA6 regulates mitochondrial function in TNBC cells.

### 3.3. Reduced ATP Production following AnxA6 Downregulation in TNBC Cells Is Associated with Metabolic Reprogramming in Favor of Fatty Acid Metabolism

Cancer cells can utilize their fatty acids for an energy source through fatty acid oxidation. We have previously shown that fatty acid-binding protein 4 (FABP4) was significantly upregulated following AnxA6 downregulation in BT-549 cells, and that this was consistent with increased cell proliferation and early onset and rapid xenograft tumor growth in mice [34]. The observations in Figure 1 and Figure 2 prompted us to speculate that the rapid growth of TNBC cells with reduced expression of AnxA6 may be dependent on the specific modulation of lipid metabolism. To test this, we first compared the lipid droplet content of our model basal-like and mesenchymal-like TNBC cell models and analyzed the effect of AnxA6 knockdown on lipid droplet (LD) abundance in the mesenchymal-like TNBC cells. We found that LDs were remarkably more abundant in the AnxA6-expressing BT-549 cells compared to the AnxA6-low MDA-468 cells (Figure 4A). The knockdown of AnxA6 expression in the BT-549 cells strongly decreased the size and number of LDs in this TNBC cell line (Figure 4B,C).

To provide further evidence on whether AnxA6 influences the LD accumulation in these cells, we evaluated the fatty acid uptake in control AnxA6-expressing (NSC) cells and following AnxA6-downregulation (A6sh5) in BT-549 cells. This analysis revealed that the fatty acid uptake significantly increased (*p* = 0.000012) in the AnxA6-downregulated cells compared to the AnxA6-expressing control cells (Figure 4D). As expected, the inhibition of carnitine palmitoyltransferase-1 (CPT1) using Etomoxir attenuated fatty acid uptake into the mitochondria in both the AnxA6-expressing and AnxA6-downregulated BT-549 cells (Figure 4D). However, the uptake of fatty acids into the mitochondria of the AnxA6-downregulated cells was significantly higher than into those of the AnxA6-expressing cells (Figure 4D). Lastly, the Seahorse XF Palmitate Oxidation Stress test revealed that the downregulation of AnxA6 in the TNBC cells significantly led to the rapid oxidation of palmitate, depicted by higher maximal OCR and ATP production compared to the AnxA6-expressing control cells (Figure 4E). Similar to the uptake studies, the presence of Etomoxir decreased the ability of TNBC cells to oxidize the palmitate. Together, these data suggest that AnxA6-low TNBC cells strongly rely on rapid fatty acid uptake and oxidation to sustain their high energetic demand and the associated rapid growth.

### 3.4. Lapatinib-Induced AnxA6 Expression Influences the Metabolic Adaptability of TNBC Cells

Although lapatinib and other tyrosine kinase inhibitors (TKIs) are potent inhibitors of their molecular targets, the efficacy of these class of drugs has been dismal in the treatment of certain solid tumors including TNBC. We recently reported that chronic treatment of AnxA6-low TNBC cells with EGFR-TKIs led to the upregulation of AnxA6 and accumulation of cholesterol in late endosomes as a novel mechanism of resistance to these drugs while the downregulation of AnxA6 in these cells rendered the cells sensitive to lapatinib [32]. Our data thus far suggest that, in TNBC cells, the expression of relatively high levels of AnxA6 is associated with increased dependency on glycolysis and LD accumulation (lipogenic phenotype), while the reduced expression of AnxA6 increased the dependency on fatty acid uptake and oxidation (lipolytic phenotype). We next determined whether the lapatinib-induced increase in AnxA6 expression in lapatinib-resistant (Lap-R) MDA-MB-468 cells (Figure 5A,B) influences the cellular metabolic phenotypes of TNBC cells.

We showed that the control (NSC) Lap-R cells with lapatinib-induced AnxA6 expression, exhibited the lipogenic phenotype and that this was changed to the more quiescent, potentially lipolytic phenotype in the AnxA6-downregulated LaP-R cells (Figure 5C,D). The basal and maximal OCR in the AnxA6-depleted Lap-R cells decreased by up to two-fold compared to that in AnxA6-expressing Lap-R cells (Figure 5E). Consistent with these changes in energy phenotype, the mitochondria function that are depicted by proton leak and ATP production were significantly decreased in the AnxA6-downregulated Lap-R cells compared to the AnxA6-expressing Lap-R cells (Figure 5F). Together with data in Figure 2 and Figure 4, this suggests that chronic treatment with EGFR-TKIs induces AnxA6 expression and alters the metabolic phenotype to the more lipogenic, slow growing lapatinib-resistant TNBC cells. Based on these findings and previous reports, the phenotypic characteristics of AnxA6-high and AnxA6-low TNBC cells are summarized in Table 1.

### 3.5. Metabolome Profiling Shows Accumulation of Gluconeogenic Amino Acids and TCA Cycle Metabolites in AnxA6 Downregulated and Lapatinib-Resistant TNBC Cells

To further elucidate the metabolic dependencies of the relatively more aggressive AnxA6-low TNBCs and the effects of lapatinib-induced expression of AnxA6 in chronically-treated (Lap-R) cells, we assessed the residual levels of the intracellular metabolites by 1H-NMR. This analysis revealed minimal differences between the levels of cellular metabolites in the parental AnxA6-expressing BT-549 and the control BT-NSC cell lines (Figure 6A).

However, the downregulation of AnxA6 in these cells led to a significant increase in the concentrations of TCA cycle metabolites such as acetate, citrate, fumarate, and succinate (Figure 6A). Alanine, arginine, and glycine, which are the precursors for gluconeogenesis, also accumulated in the AnxA6-downregulated cells (Figure 6A). In the lapatinib-resistant cells, the heatmap (Figure 6B) shows that the concentrations of TCA cycle intermediates e.g., acetate and fumarate (Figure 6B), and gluconeogenic amino acids e.g., alanine and glycine (Figure 6B) are higher in the control LAP-R cells with lapatinib-induced levels of AnxA6 compared to the AnxA6-depleted LAP-R cells with no induction of AnxA6 expression. Interestingly, the withdrawal of lapatinib from the control LAP-R cells (NSC-LW) as well as the AnxA6-downregulated LAP-R (A6sh5-LW) led to a further reduction in the gluconeogenic precursors (Figure 6B). Of particular interest, the levels of alanine, glycine, as well as fumarate and acetate were relatively higher in the control AnxA6-expressing cells compared to those in the AnxA6-depleted cells. On the contrary, the levels of citrate and lactate were lower in AnxA6-expressing LAP-R cells compared to AnxA6-downregulated LAP-R cells and that withdrawal of lapatinib led to increased levels of these metabolites. Following lapatinib withdrawal, the levels of succinate, a major entry point into the TCA cycle from amino acid precursors were higher in the AnxA6-expressing LAP-R cells but substantially decreased in the AnxA6-downregulated LAP-R cells (Figure 6B). Taken together, these data suggest that lapatinib resistance in AnxA6-low TNBC cells may lead to the activation of cell survival mechanisms that are dependent on altered lipid metabolism and/or active gluconeogenesis for energy needs.

## 4. Discussion

In the present study, we examined whether the reduced expression of AnxA6 and/or the chronic inhibition of EGFR-induced AnxA6 upregulation in TNBC cells underlies metabolic reprogramming in phenotypically distinct TNBC cell lines. Our data suggest that the mostly basal-like TNBC cell lines with reduced expression of AnxA6 strongly rely on fatty acid oxidation and gluconeogenesis to sustain their rapid growth. On the contrary, the growth of AnxA6-high, mostly mesenchymal-like TNBC cells is strongly impacted by the inhibition of glycolysis. The EGFR-TKI-induced AnxA6 expression, on the other hand, reduces the dependency of these cells on fatty acid oxidation to support slower growth and consequently, survival of the AnxA6-low TNBC cells during chronic treatment with this class of drugs. Taken together, our data suggest that the reduced expression of AnxA6 in TNBC cells is accompanied by metabolic adaptations that favor of fatty acid metabolism and provide novel insights into not only the failure of EGFR-targeted therapies as therapeutic options for TNBC, but also suggest AnxA6 as a biomarker for therapeutic intervention and/or metabolic subtyping of TNBC subsets.

Much effort has been devoted over the past decade in extensively highlighting the tumorigenic properties of AnxA6 in several cancer types [33,37], as well as in the resistance of TNBC to EGFR-TKIs [28,32,34,35,38]. In a recent study, we reported that the reciprocal expression of AnxA6 and Ras-specific guanine nucleotide-releasing factor 2 (RasGRF2) can discriminate basal-like (AnxA6-low) from mesenchymal-like (AnxA6-high) subsets of TNBC, which can help predict patient responses to chemotherapy [33]. However, the role of AnxA6 in the metabolic plasticity of TNBC cells remains unclear. Using representative basal-like and mesenchymal-like TNBC cell lines that were identified in Korolkova et al. [33], we demonstrate that AnxA6-expressing TNBC cell lines, e.g., BT-549, are less energetic and dependent on glycolysis while AnxA6-low TNBC cells, e.g., MDA-468, are more energetic and dependent on fatty acid oxidation. However, the downregulation of AnxA6 in these TNBC cell models paradoxically altered their metabolic phenotypes to the more quiescent phenotypes. Interestingly, the difference in the metabolic capacity of AnxA6-downregulated cells is attributed to the rapid uptake and degradation of fatty acids, and that in vitro culture with limiting lipid supply may rapidly render these cells with depleted lipid stores and metabolically quiescent. It is also possible that, in basal-like TNBC, the AnxA6 expression status that is dependent metabolic adaptation could be a compensatory mechanism to promote dependency on glycolysis or as suggested in this study, lipid uptake and fatty acid oxidation under defined physiological conditions including hypoxia. This notwithstanding, our data strongly suggests that AnxA6 expression status is associated with phenotypic, molecular, and metabolic heterogeneity of TNBC.

The molecular and phenotypic heterogeneity of TNBC has been amply reported [18,19], but the metabolic heterogeneity of TNBC is still poorly understood. A recent study, however, identified three metabolic-pathway subtypes (MPS) of TNBCs including the lipogenic (MPS1), the glycolytic (MPS2), and the mixed (MPS3) subgroups [39]. Consistent with this report, it is plausible to suggest that the metabolic profile of AnxA6-high mesenchymal-like TNBC cells is similar to the MSP1 subgroup. We, however, also showed that these cells are dependent on glycolysis. On the other hand, the association of the metabolic profile of the MPS2 subgroup [39] is not consistent with our study as AnxA6-low TNBC cells were more resistant to the inhibition of glycolysis and that these cells exhibited a higher dependence on fatty acid uptake and degradation. Together, these observations of heterogeneity highlight the challenges that are associated with the molecular and metabolic profiling of TNBC cell lines for studies on therapeutic intervention.

As a hallmark in TNBC progression, metabolic reprogramming enables survival and rapid proliferation in environments that are highly nutrient-deprived, such as the tumor stroma [24,40]. The remarkable difference in the mitochondrial stress test of control AnxA6-expressing versus the AnxA6-downregulated TNBC cells suggests that AnxA6 plays a critical role in the metabolic adaptation of TNBC cells during stress conditions. This is consistent with other reports suggesting diverse expression patterns for key glycolytic and mitochondrial proteins in TNBC patients’ tissues [41], and metabolic heterogeneity of non-TNBC versus TNBC cell lines based on differences in the OCR/ECAR ratio during pathway perturbation [25]. Also, compared to other breast cancer subtypes, defective mitochondria are a common characteristic of TNBC tumors [42], which may account for the differential OCR values that were observed in our panel of TNBC cell lines. Prior reports have also suggested that AnxA6-depletion alters mitochondrial respiration in other cell types [29]. Contrary to a recent study showing the rapid growth of AnxA6-downregulated cell lines in vitro and in vivo [34], serum-starvation of AnxA6-downregulated TNBC cell lines surprisingly led to reduced ATP production and proton leak suggesting a more quiescent energy phenotype. However, further analyses demonstrated that the decrease in ECAR, OCR, and ATP production in AnxA6-downregulated cells was due to the rapid uptake and degradation of fatty acids. The associated decrease in the levels of lipid stores may be sufficient to render the AnxA6-downregulated TNBC cells metabolically quiescent under the limiting supply of lipids in the cell culture system. This is consistent with rapid fatty acid uptake and dependency on rapid fatty acid oxidation as well as with the undetectable or remarkably lower lipid droplet levels in these TNBC cells that are depicted in Figure 7.

Our data are also supported by a previous report showing that AnxA6-downregulation in TNBC cells led to a significant increase in the expression of fatty acid-binding protein-4 (FABP4), which facilitates fatty acid entry into cells and/or mitochondria [34]. This is also supported by our data showing the differential sensitivity of AnxA6-expressing and AnxA6-low TNBC cells to inhibition of glycolysis and the ETC. Together, our data suggest that AnxA6-low TNBC cells are more lipolytic with rapid fatty acid uptake and oxidation to sustain their rapid proliferative rate and/or metabolic stress, while AnxA6-expressing cells are glycolytic/lipogenic. These data are of interest in terms of therapeutic strategies, as it brings attention to the need for novel TNBC subset-specific strategies to eradicate TNBC growth via the targeting of fatty acid uptake mechanisms and/or subsequently oxidation of fatty acids in basal-like and mesenchymal-like TNBCs.

Perhaps the most interesting observation in this study is identifying the metabolic dependencies of lapatinib-resistant AnxA6-low TNBC cells. Despite the development of several TKIs against EGFR and/or other ErbB receptors, the dismal response and development of resistance against this class of drugs, including the third generation TKIs, has limited their use for the management of TNBC [9,14,15,17,32]. However, supportive evidence for their failure as therapeutic agents for TNBC requires novel and more impactful strategies including a better understanding of acquired resistance to this class of drugs. Amongst the EGFR-TKI family of drugs, we have shown that chronic treatment of TNBC cells with lapatinib significantly induces the expression of AnxA6, cholesterol accumulation, and the activation of the MAP kinase pathway as an adaptive response to the drug treatment [32]. Our findings in this study revealed that, besides sensitivity to the inhibition of glycolysis, the metabolic phenotype of lapatinib-resistant (Lap-R) AnxA6-expressing TNBC cells is significantly altered from energetic/lipogenic to quiescent/lipolytic phenotype following AnxA6-downregulation. These data strongly support the notion that the increase in AnxA6 expression following chronic TKI treatment is an adaptive mechanism by the potentially lipolytic AnxA6-low TNBC cells to reprogram their metabolic responses under chronic stress conditions towards the glycolytic phenotype with more mesenchymal-like but reduced growth properties. This is supported by previous reports suggesting that treatment is associated with the ER stress [43,44]. Other studies have shown that lapatinib treatment is associated with an extensive phosphorylation-mediated reprogramming of cellular glycolytic activity, widespread changes of corresponding metabolites, and an increased sensitivity towards the inhibition of glycolysis [1]. While these studies were limited to glucose metabolism, our study expands these observations to lipid metabolism in basal-like TNBCs.

In addition to building proteins, amino acids are instrumental as important energy sources [45]. Our ^1^H-NMR metabolome analysis identified several gluconeogenic amino acids as additional metabolic precursors that were associated with the survival of TNBC cell lines following AnxA6-downregulation or chronic lapatinib treatment. The accumulation of alanine, glycine, and other gluconeogenic amino acids in AnxA6-downregulated TNBC cells supports the impairment of gluconeogenesis in favor of the rapid uptake and oxidation of fatty acids. This is consistent with a recent report suggesting that the reduced expression of AnxA6 in adipocytes underlies their ability to impair fat storage and adiponectin release during oxidative stress [30]. Another study demonstrated that hepatic gluconeogenesis was not sufficiently attenuated in AnxA6-KO mice despite normal hepatocytes, insulin levels/signaling, and the expression profiles of insulin-sensitive transcription factors [31,46,47]. Interestingly, the upregulation of glycine metabolism has been shown to correlate with high tumor cell proliferation and poor prognosis in patients [48], while arginine is involved in immunity and tolerance that is commonly dysregulated in malignant melanoma, renal cell carcinoma, and prostate cancer [49]. In this context, the increase in amino acid metabolites in response to AnxA6 depletion, and the decease of these metabolites following lapatinib resistance are highly suggestive of the reliance of the basal-like AnxA6-low TNBC cells on fatty acid metabolism during chronic drug treatment and/or sustained cellular stress conditions.

## 5. Conclusions

In summary, our data reveal that the downregulation of AnxA6 is associated with metabolic adaptation towards fatty acid degradation, and that acquired resistance of AnxA6-low basal-like TNBC cells to EGFR-targeted therapies is associated with the glycolytic/lipogenic metabolic phenotype. This study, therefore, provides additional insights into the metabolic heterogeneity in TNBC and suggests AnxA6 as a potential biomarker for not only therapeutic intervention but also the metabolic subtyping of TNBC subsets. However, given the paradoxical quiescent metabolic phenotype following AnxA6-downregulation in TNBC cells, metabolic phenotyping of certain cancer cells in vitro should be interpreted with caution due to the rapid depletion of the limiting lipid supply in the culture media.

## Figures and Tables

**Figure 1 cancers-14-01108-f001:**
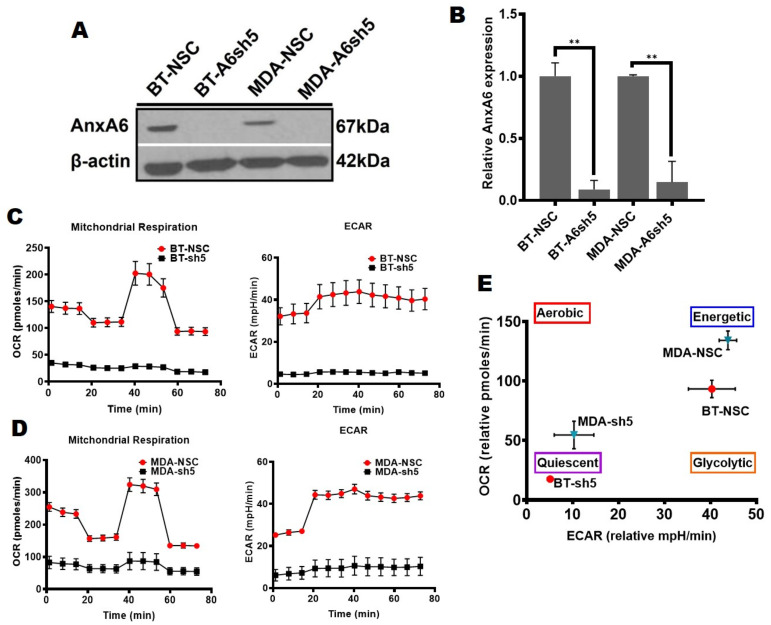
Metabolic heterogeneity of TNBC cells is AnxA6-dependent. (**A**,**B**) AnxA6 downregulation in the representative basal-like (MDA-468) and mesenchymal-like (BT-549) TNBC cells. The expression levels of AnxA6 were assessed by Western blotting of whole cell extracts from the control (NSC) and AnxA6-downregulated (A6sh5) BT-459 and MDA-468 cells. The detection of β-actin was used as a loading control. (**B**) Densitometric analysis of AnxA6 was carried out using ImageJ and expressed as the intensity of each protein band normalized to β-actin. (**C**,**D**) The mitochondrial stress test was used to obtain bioenergetic parameters of the control and AnxA6-downregulated BT-549 and MDA-468 TNBC cells. Real-time oxygen consumption rates (OCR; pmoles/min/cell count) and the extracellular acidification rates (ECAR; mpH/min/cell count) measured over time (min) for BT-549 (**C**) and MDA-468 (**D**) AnxA6-expressing and -downregulated cell lines. (**E**) The maximal OCR/ECAR ratio were used to plot the energy phenotype of the control and AnxA6 downregulated BT-549 and MDA-468 TNBC cells following the mitochondrial stress test. All data are presented as the mean ± SEM, *n* = 3. ** denotes *p* ≤ 0.01.

**Figure 2 cancers-14-01108-f002:**
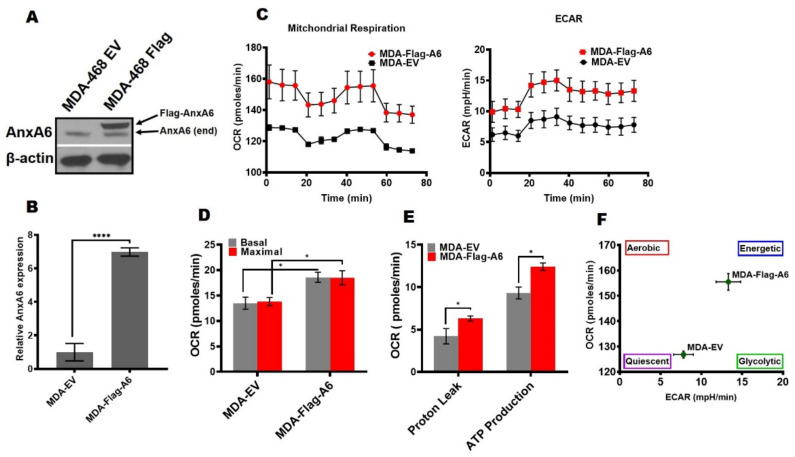
Response of cellular bioenergetics for the overexpression of AnxA6 in AnxA6-low MDA-468. (**A**) AnxA6 overexpression in the representative basal-like MDA-468 TNBC cell lines. The expression levels of AnxA6 were assessed by Western blotting of whole cell extracts from control (EV) and AnxA6-overexpressing (Flag-A6) MDA-468 cells. The detection of β–actin was used as a loading control. (**B**) Densitometric analysis of AnxA6 was carried out using ImageJ and expressed as the intensity of each protein band normalized to β-actin (**C**) The mitochondrial stress test was used to obtain bioenergetic parameters on EV and Flag-A6 MDA-468 cells. Real-time oxygen consumption rates (OCR; pmoles/min/cell count) and extracellular acidification rates (ECAR; mpH/min/cell count) measured over time (min). (**D**) Basal and maximal OCR rates; (**E**) Proton leak and ATP production rates. (**F**) The maximal OCR/ECAR ratio were used to plot the energy phenotype of the control (EV) and overexpression (Flag-A6) MDA-468 cell lines. All the data presented as the mean ± SEM, *n* = 3. * denotes *p* ≤ 0.05; **** denotes *p* ≤ 0.0001 by two-way ANOVA.

**Figure 3 cancers-14-01108-f003:**
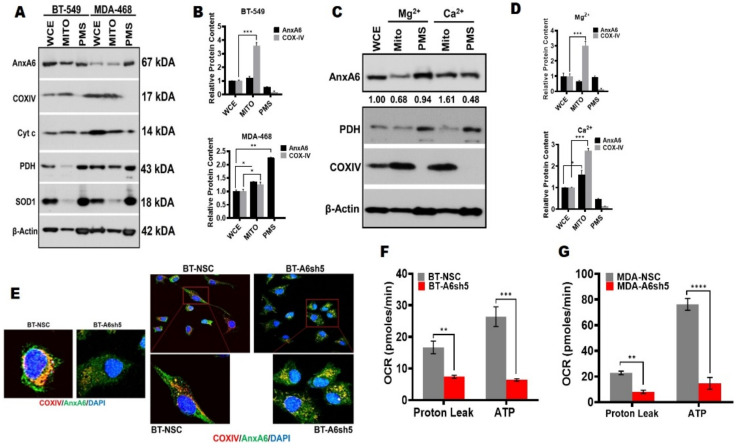
Downregulation of AnxA6 alters the mitochondrial function. (**A**,**B**) Localization of AnxA6 in mitochondria. Crude mitochondria were isolated from parental BT-549 and MDA-468 TNBC cells and the expression of AnxA6 and the indicated proteins were assessed by Western blotting. The detection of β–actin was used as a loading control. (**C**,**D**) The effects of divalent ions on localization of AnxA6 to mitochondria. Mitochondria-enriched factions were prepared in the presence of Mg^2+^ or Ca^2+^ and analyzed as in (**A**) above. Densitometry analysis of AnxA6 was carried out using ImageJ and expressed as the intensity of each protein band normalized to β-actin. (**E**) Co-localization of AnxA6 and COXIV in the AnxA6-expressing control BT-NSC and AnxA6-downregulated BT-A6sh5 cells. Images were acquired using the ×40 objective. (**F**,**G**) The effect of AnxA6-downregulation in BT-549 and MDA-468 TNBC cells on proton leak and overall ATP production (ATP). WCE: whole cell extracts, MITO: mitochondrial fraction; PMS: post-mitochondrial supernatants. All the data presented as the mean ± SEM, *n* = 3. * denotes *p* ≤ 0.05; ** denotes *p* ≤ 0.01; *** denotes *p* ≤ 0.001; **** denotes *p* ≤ 0.0001) by two-way ANOVA.

**Figure 4 cancers-14-01108-f004:**
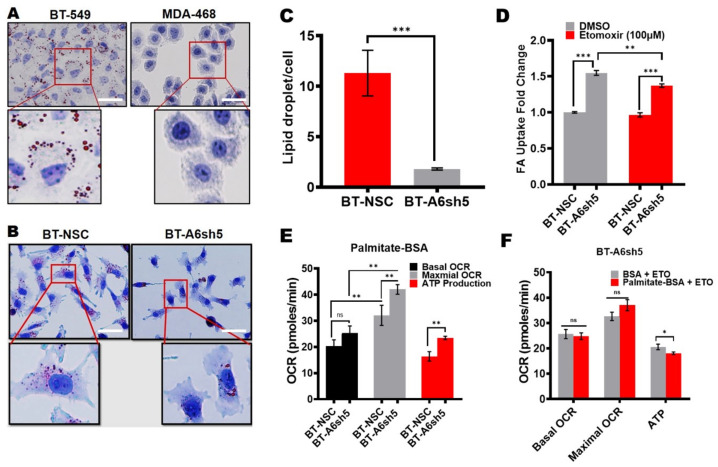
The reduced expression of AnxA6 in TNBC cells is associated with changes in lipid metabolism. (**A**) Detection of lipid droplets in the BT-549 and MDA-468 parental TNBC cell lines. (**B**) Detection of lipid droplets in the control and AnxA6-downregulated BT-549 cell lines. The cells were grown for 24 h on coverslips and stained with Oil Red O as described in the methods section. Scale bars represent 20 µm (**C**) Quantification of lipid droplets by using Image-J software. (**D**) The measurement of fatty acid uptake in the control and AnxA6-downregulated BT-549 cells. Fatty acid uptake was performed in the presence and absence of Etomoxir (100 µM). (**E**,**F**) The measurement of fatty acid oxidation. the mitochondria stress test was performed in the control and AnxA6-downregulated BT-549 cells in the presence of carrier BSA or the fatty oxidation substrate palmitate-BSA and either untreated or treated with Etomoxir. Cells were imaged on a Keyence BZ-X800 fluorescence microscope (Keyence Corp.). A total of four fields of each condition were used for the quantification of the Oil red-O staining using the NIH ImageJ software All the data presented as the mean ± SEM, *n* = 3. ns denotes *p > 0.5;* * denotes *p* ≤ 0.05; ** denotes *p* ≤ 0.01; *** denotes *p* ≤ 0.001 by two-way ANOVA.

**Figure 5 cancers-14-01108-f005:**
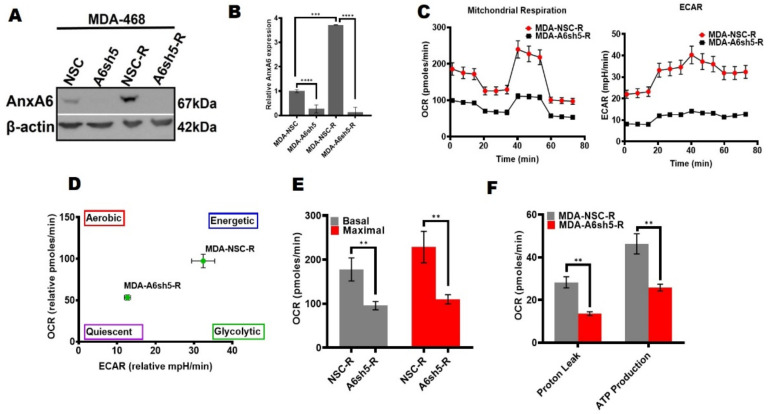
Lapatinib-induced AnxA6 expression is associated with the metabolic adaptation of TNBC cells. (**A**) Analysis of AnxA6 expression in the control and lapatinib-resistant AnxA6-expressing (NSC) and AnxA6-downregulated MDA-468 cells by Western blotting. The detection of β-actin was used as the loading control. (**B**) Densitometry analysis of AnxA6 was carried out using ImageJ and expressed as the intensity of each protein band normalized to β-actin. (**C**–**F**) The effect of AnxA6 downregulation on the metabolic activity of lapatinib-resistant MDA-468 cells. Mitochondrial respiration in the control and lapatinib-resistant AnxA6-expressing (NSC) and AnxA6-downregulated (A6sh5) MDA-468 cells was assessed by a mitochondrial stress test (**C**), and the energy phenotype (**D**), basal and maximal OCR (**E**), and proton leak, and overall ATP production (**F**) were obtained from Seahorse XF analysis. All the data presented as the mean ± SEM, *n* = 3. ** denotes *p* ≤ 0.01; denotes *** *p* ≤ 0.001; **** denotes *p* ≤ 0.0001 by one-way ANOVA.

**Figure 6 cancers-14-01108-f006:**
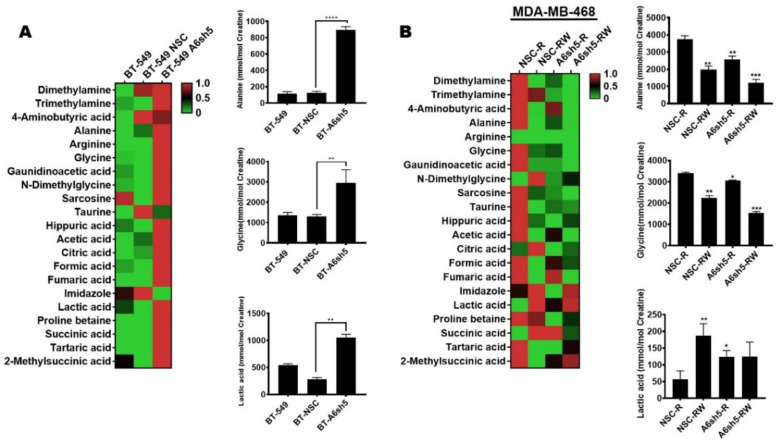
Metabolic dependencies of AnxA6-low and lapatinib-resistant TNBC cells. Metabolites were extracted from the indicated subclones of BT-549, and the MDA-468 cells were analyzed by H1-NMR. The heat-map shows the normalized enriched scores and the levels of the indicated metabolites for the control and AnxA6 downregulated BT-549 cells (**A**) or lapatinib-resistant MDA-468 cells that were cultured either in continuous lapatinib treatment (R) or in the absence of lapatinib (RW). Each bar represents the average concentration of the indicated metabolite from two biological replicates. All the data presented as the mean ± SEM, *n* = 3. Asterisks (* *p* ≤ 0.05; ** *p* ≤ 0.01; *** *p* ≤ 0.001; **** *p* ≤ 0.0001) are normalized to NSC (**A**) and NSCR-R (**B**). *p* values were analyzed by a one-way ANOVA test.

**Figure 7 cancers-14-01108-f007:**
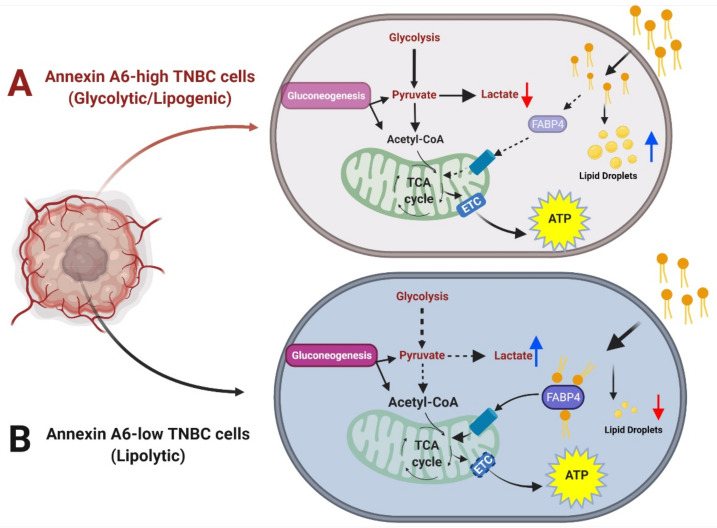
Schematic diagram depicting the effect of reduced AnxA6 expression on the metabolic adaptation of TNBC cells. AnxA6-high TNBC cells exhibit high glycolytic output and increased deposition of lipid droplets suggesting a glycolytic/lipogenic phenotype. AnxA6-low TNBC cells on the other hand, exhibit low glycolysis but high lipid turnover suggesting a lipolytic phenotype. (**A**) Annexin A6-high TNBC cells that exhibit the glycolytic/lipogenic phenotype; (**B**) Annexin A6-low TNBC cells that exhibit the lypolytic phenotype.

**Table 1 cancers-14-01108-t001:** Phenotypic characteristics of AnxA6-high and AnxA6-low TNBC cells.

Property	AnxA6-High *	AnxA6-Low
Cellular morphology	Mesenchymal-like	Basal-like
Cell proliferation	Low	High
Cell migration/invasion	High	Low
Lapatinib sensitivity	Resistant	Sensitive
Maximal OCR	High	Low
Maximal ECAR	High	Low
ATP production	High	Low
Fatty acid oxidation	Slow	Rapid
Lipid droplets	Abundant	Undetected to low
Metabolic phenotype	Glycolytic/Lipogenic	Lipolytic

* AnxA6-high TNBC cells: TNBC cells naturally expressing relatively high levels of AnxA6; AnxA6-low cells: TNBC cells expressing relatively low levels of AnxA6 and/or AnxA6 downregulation in AnxA6-high TNBC cells.

## Data Availability

The data presented in this study are available on request from the corresponding author.

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
