# Peer review of "Reduced Expression of Annexin A6 Induces Metabolic Reprogramming That Favors Rapid Fatty Acid Oxidation in Triple-Negative Breast Cancer Cells"

_cancers, 2022, doi:10.3390/cancers14051108_

Round 1
Reviewer 1 Report
Williams et. al. studied the effect of Annexin A6 (AnxA6) on triple-negative breast cancer cell metabolism. The paper, however, is very confusing. Authors have demonstrated that AnxA6 is critical for cell metabolism as shown by the mitochondrial stress test. Higher AnxA6 led to a more energetic phenotype while AnxA6 knockdown led to a quiescent phenotype. On the contrary, natively AnnA6-low cells like MDA-468 are energetic and Ann6-high like MDA-231 are quiescent which makes it confusing.
In the introduction section, the second paragraph, MDA-MB-468 was introduced as cells with low OXPHOS then mentioned together with BSL-TNBC like cells that have more active metabolism.
In Figure 4A, AnxA6 like SOD1 and PDH were in post-mitochondrial supernatants while authors indicate that it is mainly associated with mitochondria.
In Figure 6, there is no comparison of mitochondrial respiration in lapatinib resistant and lapatinib sensitivity cells.
I see that the paper only shows the critical role of AnxA6 in the mitochondrial OXPHOS and TCA cycle as shown by OCR/ECAR and amino acids related to that. This, however, does not explain the heterogeneity of TNbC cells but probably it could be a compensatory mechanism to promote defective OXPHOS in mesenchymal-like cells. Therefore, I suggest that results in sections 3.1 and 3.2 should be removed or could be introduced at the end.
Author Response
Reviewer #1
Williams et. al. studied the effect of Annexin A6 (AnxA6) on triple-negative breast cancer cell metabolism. The paper, however, is very confusing. Authors have demonstrated that AnxA6 is critical for cell metabolism as shown by the mitochondrial stress test. Higher AnxA6 led to a more energetic phenotype while AnxA6 knockdown led to a quiescent phenotype. On the contrary, natively AnnA6-low cells like MDA-468 are energetic and Ann6-high like MDA-231 are quiescent which makes it confusing.
Response: We thank the reviewer for taking time out to critically read our manuscript and to provide very insightful comments that overall have improved our report.
We sincerely appreciate the reviewer’s concern on the metabolic phenotypes of parental TNBC versus the effect of AnxA6 down regulation in TNBC cells. Figure 1F is indeed a replication of a previous report (Ref. # 26) even though different AnxA6-high cell lines were tested in our study. The effect of AnxA6 depletion in the AnxA6-low MDA-468 and the AnxA6-high BT-549 is essentially similar, and this is opposite as expected, when AnxA6 was up regulated in MDA-468 cells. While this unequivocally suggest a role of this tumor suppressor in metabolic reprogramming of TNBC cells, we have adhered to the reviewer’s recommendation to remove section 3.1 and 3.2 on metabolic heterogeneity of TNBC cells (see below). We have accordingly, edited the sections regarding this phenomenon in the revised manuscript.
In the introduction section, the second paragraph, MDA-MB-468 was introduced as cells with low OXPHOS then mentioned together with BSL-TNBC like cells that have more active metabolism.
Response: We thank the reviewer for this significant oversight on our part. Indeed, MDA-468 TNBC cells are among the best characterized as basal-like TNBC cells. We have deleted this statement in the revised manuscript (line 59).
In Figure 4A, AnxA6 like SOD1 and PDH were in post-mitochondrial supernatants while authors indicate that it is mainly associated with mitochondria.
Response: As a predominantly cytosolic protein, and like most Annexins, AnxA6 associates with cellular membranes in a Ca2+ dependent manner. Based on the method used to isolate crude mitochondria, the post mitochondria supernatant also contains crude cellular membranes and certainly contain AnxA6. Therefore, only a fraction of AnxA6 is expected to be associated with mitochondria, while the rest will be associated with plasma and other cellular membranes in the post mitochondria supernatant. The cell fractionation to isolate crude mitochondria was an additional assay to support the immunofluorescence staining of COXIV and AnxA6. We have also edited the statement to reflect the distribution of AnxA6 in several membrane bound cellular organelles.
In Figure 6, there is no comparison of mitochondrial respiration in lapatinib resistant and lapatinib sensitivity cells.
Response: We thank the reviewer for this very important concern. In our previous studies, we showed that AnxA6 expressing TNBC cells are refractory to EGFR TKI treatment and that down regulation of AnxA6 sensitized the cells to these drugs (Koumangoye, 2013, Ref# 38). Our analysis includes the Lapatinib resistant control AnxA6-low MDA-468 cells that in the presence of chronic lapatinib treatment leads to AnxA6 upregulation and the development of resistance; as well as Lapatinib resistant AnxA6 down regulated MDA-468 cells in which lapatinib induced expression of AnxA6 is blocked and these cells remain sensitive to lapatinib (Widatalla, 2019, Ref.# 35 ). These two subclones therefore, represent isogenic lapatinib resistant and lapatinib sensitive TNBC cells.
I see that the paper only shows the critical role of AnxA6 in the mitochondrial OXPHOS and TCA cycle as shown by OCR/ECAR and amino acids related to that. This, however, does not explain the heterogeneity of TNbC cells but probably it could be a compensatory mechanism to promote defective OXPHOS in mesenchymal-like cells. Therefore, I suggest that results in sections 3.1 and 3.2 should be removed or could be introduced at the end.
Response: We completely agree with the reviewer regarding the possibility that the AnxA6 expression status dependent changes in amino acid metabolism and the OCR/ECAR could be a compensatory mechanism to promote defective metabolic output depicted by dependency on glycolysis or as suggested in this study, lipid uptake and fatty acid oxidation. We have revised the discussion (lines 501-505), to include this very important inference.
Regarding the placement of sections 3.1 and 3.2 of the manuscript, our thought process was to provide a picture of the heterogeneity of metabolic differences between AnxA6-high and AnxA6-low TNBC cells and then use RNAi technology and overexpression of AnxA6 to validate the role of AnxA6 as a driver for the metabolic adaptation of these cells. Given that a previous report amply studied metabolic heterogeneity of similar TNBC cells (Ref# 26), the ambiguity introduced in our data, and as recommended by this reviewer, we have deleted these sections from the revised manuscript.

Reviewer 2 Report
Thank you for doing this on TNBC.
Line 19
…proliferative and invasive TNBC… Consider change to …..distinct metabolic adaptations in TNBC subsets in response to…..
This is confusing, TNBC is considered both invasive and proliferateive (that’s the problem)
Please clarify your definition of invasive TNBC and proliferative TNBC
Lin24
…rapidly growing (AnxA6-low) versus invasive (AnxA6-high)….
There are trippel negative DCIS. However, when writing about TNBC in this context is understood that it has invasive potential even being proliferative. Its mor or less not either.
You might have meant versus slow growing (AnxA6-high)
Line 34
…adaption in TNBC (don’t write proliferative and invasive, unless you specify this. If you specify! parts of chapter 3 Results line 246 and 247 should be clarified in abstract and moved to introduction.
Line 48
Meaning of EGFR
Line 92
…..rapidly growing versus MORE invasive TNBC cells, ….
- Results
Line 243 to 250 is introduction.
You might move this to introduction and write something like. “ In this analysis we found low expression of AnxA6 in BCL-TNBC (MDA-468, HCC-70) and high expression of AnxA6 in MES-TNBC (MDA-23, BT-549) se Fig.1A and 1B. This is well in hand with our previous report.
Line 324
AnxA6 in …. is associated with, something missing?
You have several findings in this and previous papers. You might make a table to summarize all these findings (It is obvious for you). Makes it easier for the reader to follow.
|
|
Cell proliferation |
ATP production |
Mitocondrial production |
Lipid droplets |
|
|
|
Anx6 high |
|
|
|
|
|
|
|
Anx6 low |
|
|
|
|
|
|
Author Response
Reviewer #2
Thank you for doing this on TNBC.
Line 19
…proliferative and invasive TNBC… Consider change to …..distinct metabolic adaptations in TNBC subsets in response to…..
This is confusing, TNBC is considered both invasive and proliferateive (that’s the problem)
Please clarify your definition of invasive TNBC and proliferative TNBC
Response: We agree with the reviewer that the use of AnxA6-high versus AnxA6-low; mesenchymal-like versus basal-like and invasive versus proliferative may be confusing. Based on the reviewer’s comment, we have simplified this to “basal-like or AnxA6-low” versus “mesenchymal-like or AnxA6-high” cells, as this is consistent with the descriptions in Lehmann, BD et al., 2011 and in our recent study, Korolkova OY et al., 2020, Plos One).
Lin24
…rapidly growing (AnxA6-low) versus invasive (AnxA6-high)….
There are trippel negative DCIS. However, when writing about TNBC in this context is understood that it has invasive potential even being proliferative. Its mor or less not either.
You might have meant versus slow growing (AnxA6-high)
Response: We agree with the reviewer and have used a more consistent terminology
Line 34
…adaption in TNBC (don’t write proliferative and invasive, unless you specify this. If you specify! parts of chapter 3 Results line 246 and 247 should be clarified in abstract and moved to introduction.
Response: We agree with the reviewer and have used a more consistent terminology
Line 48
Meaning of EGFR
Response: We thank the reviewer for the omission. We have now defined “Epidermal Growth Factor Receptor” (EGFR) in the revised manuscript.
Line 92
…..rapidly growing versus MORE invasive TNBC cells, ….
Response: We agree with the reviewer and have used a more consistent terminology.
- Results
Line 243 to 250 is introduction.
You might move this to introduction and write something like. “ In this analysis we found low expression of AnxA6 in BCL-TNBC (MDA-468, HCC-70) and high expression of AnxA6 in MES-TNBC (MDA-23, BT-549) se Fig.1A and 1B. This is well in hand with our previous report.
Response: We agree with the reviewer on the introductory comments in the Results section. In some sections, a preamble is necessary to provide context to the experiments and the ensuing data. However, on recommendation from Reviewer #1, this section has be deleted from the manuscript.
Line 324
AnxA6 in …. is associated with, something missing?
Response: We thank the reviewer for this comment, and we have edited the sentence to read “…AnxA6 is associated with,…”
You have several findings in this and previous papers. You might make a table to summarize all these findings (It is obvious for you). Makes it easier for the reader to follow.
|
|
Cell proliferation |
ATP production |
Mitocondrial production |
Lipid droplets |
|
|
|
Anx6 high |
|
|
|
|
|
|
|
Anx6 low |
Response: We thank the reviewer, and have included “Table 1. Phenotypic characteristics of AnxA6-high and AnxA6-low TNBC cells” which includes additional phenotypic characteristics; and new Fig. 7, a schematic diagram depicting the metabolic changes associated with altered expression of AnxA6 in TNBC cells, which together provides additional clarity to our findings.

Reviewer 3 Report
The manuscript “Reduced expression of Annexin A6 induces metabolic reprogramming that favours rapid fatty acid oxidation in triple-negative breast cancer cells” by Williams SD et al. is a very interesting research article focused on the metabolic changes affected by Annexin A6 levels in triple negative breast cancer.
The manuscript is well structured and the data are well presented even if some points have to be addressed so that the conclusion may be better supported by the results.
-In the figure 1F, the authors reported a graphical representation of the energy phenotype based on OCR/ECAR ratios for all selected cell lines. In particular, this analysis indicated that the AnxA6 high BT-549 cells showed a more quiescent metabolic state in comparison with the proliferative AnxA6 low MDA-468 cells. These results are not in agreement with the data reported in the figure 2E, where the authors indicated that both the control MDA-468 and BT-549 cells were profiled as highly energetic phenotypes.
-In order to give force to the author’ hypothesis, partially demonstrated by the results showed in the figure 5D, E and F (low levels of AnxA6 are associated with a rapid fatty acid uptake ad oxidation) the authors should reports similar results in MDA-468 cells.
-At lines 275-279, it is showed that the Anx6-low MDA 468 are highly energetic and proliferative. This observation is not in agreement with that reported at lines 428-430, where the authors talk about a paradoxical quiescent phenotype of AnxA6-low TNBC. This point should be well addressed and clarified.
-The legend of figure 5 E-F should be better described, in particular the authors should clearly indicate in the text the meaning of “Palmitate : BSA” .
Author Response
Reviewer #3
The manuscript “Reduced expression of Annexin A6 induces metabolic reprogramming that favours rapid fatty acid oxidation in triple-negative breast cancer cells” by Williams SD et al. is a very interesting research article focused on the metabolic changes affected by Annexin A6 levels in triple negative breast cancer.
The manuscript is well structured and the data are well presented even if some points have to be addressed so that the conclusion may be better supported by the results.
Response: We are grateful for the reviewer’s input and sincerely think that the recommendations and responses to the concerns have not only strengthened but also resolved the ambiguities in our initial submission.
-In the figure 1F, the authors reported a graphical representation of the energy phenotype based on OCR/ECAR ratios for all selected cell lines. In particular, this analysis indicated that the AnxA6 high BT-549 cells showed a more quiescent metabolic state in comparison with the proliferative AnxA6 low MDA-468 cells. These results are not in agreement with the data reported in the figure 2E, where the authors indicated that both the control MDA-468 and BT-549 cells were profiled as highly energetic phenotypes.
Response: We agree with the reviewer regarding this concern, which was also indicated by Reviewer #1. Although the metabolic phenotypes of AnxA6-low and of AnxA6-high TNBC cell lines in our study agree with a previous report (Ref# 26), the metabolic phenotypes of control and AnxA6 down regulated cells were surprisingly different. However, the use of isogenic MDA-468 cells in which AnxA6 is either down regulated or overexpressed strongly support a role of AnxA6 in metabolic adaptation of TNBC cells. Based on reviewer #1’s recommendation and to resolve the ambiguity, this section has been deleted in the revised manuscript and any associated text edited accordingly.
-In order to give force to the author’ hypothesis, partially demonstrated by the results showed in the figure 5D, E and F (low levels of AnxA6 are associated with a rapid fatty acid uptake ad oxidation) the authors should reports similar results in MDA-468 cells.
Response: We thank the reviewer again for requesting rigor in our study. In several sections of the manuscript, we have used MDA-468 including lapatinib resistant MDA-468 as the AnxA6-low, and BT-549 as the AnxA6-high TNBC cells. We did not anticipate significant changes in Fatty acid uptake and degradation in MDA-468 cells based on our findings that lipid droplets were undetected in both control and following altered expression of AnxA6 in MDA-468 cells. However, the abundant lipid droplets in BT-549 and drastic depletion following AnxA6 down regulation in these AnxA6-high cells, guided our decision to use control and AnxA6 depleted BT-549 cells as the best model to test the concept.
-At lines 275-279, it is showed that the Anx6-low MDA 468 are highly energetic and proliferative. This observation is not in agreement with that reported at lines 428-430, where the authors talk about a paradoxical quiescent phenotype of AnxA6-low TNBC. This point should be well addressed and clarified.
Response: We thank the reviewer for recognizing this concern which was also indicated by Reviewer #1. Based on reviewer# 1’s recommendation and to resolve this ambiguity, this section has been deleted in the revised manuscript and any associated text edited accordingly.
-The legend of figure 5 E-F should be better described, in particular the authors should clearly indicate in the text the meaning of “Palmitate : BSA” .
Response: We thank the reviewer for this oversight. In this figure, Palmnitate-BSA is written in error as Palmitate:BSA. We have corrected this in the revised manuscript and clearly indicated BSA as the palmitate carrier, and BSA-conjugated palmitate (palmitate-BSA) as the fatty acid oxidation substrate in the text (Section 2.10, Line 218) and the legend of new Fig. 4 (Lines 367-368).

Round 2
Reviewer 1 Report
Manuscript is better now. However, still minor changes still need to be done.
Page 1, line 13, In the Simple abstract: Please remove the word Paradoxically, as you did not mention anything (in the abstract) related to Mesenchymal cells being quiescent while basal-like being energetic. so it would be inappropriate here (the reader won't understand).
Abstract, page 1, Line 27, please move paradoxically to the next sentence and you can then explain that “paradoxically, mesenchymal cells were quiescent while basal-like were energetic. This can be added after "To explain this paradox…."
Introduction: Page 2, Line 82, the sentence starting with “recent studies …” is out of context
Page2, Line 95 "adaptations for TNBC cells to stress. Furthermore, we show that basal-like have AnxA6-low and mesenchymal-like cells have AnxA6-high status. This can be used in the future as a metabolic biomarker for subtyping of TNBC subsets.
Results: Line 350, it is not clear for the reader what is the “paradoxical effects seen in Fig 1 and 2” You might want to explain that although mesenchymal cells are quiescent, they are Anx 6 high while basal-like are energetic while being Anx 6 low"
Page 9, Line 388, This sentence should be in the discussion as the authors did not compare the growth of mesenchymal-like cells in vitro and in vivo but they rather rely on previously reported findings.
Author Response
Reviewer # 1 Round -2
Manuscript is better now. However, still minor changes still need to be done.
We sincerely appreciate the attention to detail of this reviewer, and grateful for the tremendous time and energy input into critically reading the revised version of the manuscript. Below are our responses to the specific comments.
Page 1, line 13, In the Simple abstract: Please remove the word Paradoxically, as you did not mention anything (in the abstract) related to Mesenchymal cells being quiescent while basal-like being energetic. so it would be inappropriate here (the reader won't understand).
Response: We agree with the reviewer and have deleted the word “paradoxically”.
Abstract, page 1, Line 27, please move paradoxically to the next sentence and you can then explain that “paradoxically, mesenchymal cells were quiescent while basal-like were energetic. This can be added after "To explain this paradox…."
Response: We thank the reviewer for this oversight and have also, revised this section of the abstract.
Introduction: Page 2, Line 82, the sentence starting with “recent studies …” is out of context
Response: This paragraph is a follow-up of the previous paragraph devoted to introduction of AnxA6 and energy metabolism. This section of the introduction is intended to emphasize the effect of AnxA6 expression status in TNBC subtypes (Basal-like versus mesenchymal-like). For clarity we have revised the sentence structure by taking out the word “also” which made it redundant.
Page2, Line 95 "adaptations for TNBC cells to stress. Furthermore, we show that basal-like have AnxA6-low and mesenchymal-like cells have AnxA6-high status. This can be used in the future as a metabolic biomarker for subtyping of TNBC subsets.
Response: We thank the reviewer for these edits and totally agree to modify the statement as follows:
“… bioenergetic adaptations to stress. Furthermore, we show that basal-like (AnxA6-low) and mesenchymal-like (AnxA6-high) TNBCs cells exhibit distinct metabolic phenotypes, suggesting that AnxA6 expression status can be used in the future as a metabolic biomarker for subtyping of TNBC subsets.”
Results: Line 350, it is not clear for the reader what is the “paradoxical effects seen in Fig 1 and 2” You might want to explain that although mesenchymal cells are quiescent, they are Anx 6 high while basal-like are energetic while being Anx 6 low"
Response: We thank the reviewer for t critically reading our revised manuscript and identifying these inconsistencies. We have simply removed the word “paradoxical” from the sentence.
Page 9, Line 388, This sentence should be in the discussion as the authors did not compare the growth of mesenchymal-like cells in vitro and in vivo but they rather rely on previously reported findings.
Response: We also sincerely thank the reviewer for this suggestion and have deleted the statement in line 388, and completed the overall conclusion of the study as follows: “… should be interpreted with caution due to rapid depletion of the limiting lipid supply in the culture media”.

Reviewer 3 Report
The authors well addressed each points reported in the previous revision!
Author Response
The reviewer indicated that "The authors well addressed each points reported in the previous revision!"